# An ensemble deep learning model for medical image fusion with Siamese neural networks and VGG-19

Venu Allapakam[1], Yepuganti Karuna[2]*

1 School of Electronics Engineering, Vellore Institute of Technology, Vellore, India, 2 School of Electronics Engineering, VIT-AP University, Amaravathi, India

* karuna.y@vitap.ac.in

**Data Availability Statement:** All relevant image dataset are available from https://github.com/praneethMohan/GIST-CT-PET/tree/main, https://github.com/MorvanLi/image-fusion-zoom/tree/

## Abstract

Multimodal medical image fusion methods, which combine complementary information from many multi-modality medical images, are among the most important and practical approaches in numerous clinical applications. Various conventional image fusion techniques have been developed for multimodality image fusion. Complex procedures for weight map computing, fixed fusion strategy and lack of contextual understanding remain difficult in conventional and machine learning approaches, usually resulting in artefacts that degrade the image quality. This work proposes an efficient hybrid learning model for medical image fusion using pre-trained and non-pre-trained networks i.e. VGG-19 and SNN with stacking ensemble method. The model leveraging the unique capabilities of each architecture, can effectively preserve the detailed information with high visual quality, for numerous combinations of image modalities in image fusion challenges, notably improved contrast, increased resolution, and lower artefacts. Additionally, this ensemble model can be more robust in the fusion of various combinations of source images that are publicly available from Havard-Medical-Image-Fusion Datasets, GitHub. and Kaggle. Our proposed model performance is superior in terms of visual quality and performance metrics to that of the existing fusion methods in literature like PCA+DTCWT, NSCT, DWT, DTCWT+NSCT, GADCT, CNN and VGG-19.

## 1. Introduction

Medical image fusion is a hot research topic for many researchers because it holds crucial information for precious diagnosis. Most imaging modalities provide unique and unilateral information content, which often causes great inconvenience in precious diagnosis for health care professionals [1]. Doctors typically need to combine multiple types of medical images to obtain sufficient information for an accurate diagnosis. The kind of method that has the most potential for this purpose is image fusion, where simultaneous viewing of information from more than one imaging modality can be beneficial. The fused image holds detailed information with feature enhancement, Image sharpening, and improved classification can be

main/Medical_Image_ Fusion_Methods/Havard-Medical-Image-Fusion-Datasets, https://www.kaggle.com/datasets/mateuszbuda/lgg-mri-segmentation?rvi=1.

**Funding:** The author(s) received no specific funding for this work.

**Competing interests:** The authors have declared that no competing interests exist.

benefited [2]. Overall, medical image fusion plays a vital role in enabling more informed and accurate medical decisions, leading to better patient outcomes.

The main difficulty in image fusion is how to extract essential features from source images and combine them to produce the fused image. Many medical image fusion algorithms have been implemented during this past decade based on either decomposition or learning techniques. In general, multi-scale transform (MST)-based image fusion tasks are usually carried out in three essential steps: image decomposition, fusion, and reconstruction [3]. Fig 1 illustrates an example of image fusion using discrete wavelet transformations (DWT) as image decomposition, followed by weight map generation and reconstruction to get the final fused output [4].

The image transforms and fusion rule, often known as a pixel-level decision map, are the primary factors that impact fusion quality. This decision map is often built in two steps activity level measurement and weight assignment [5–7]. The absolute value of the decomposed coefficient and choose-max fusion rules were employed for activity measurement and assigning weights in many conventional transformation domains [7–10]. Activity level measurements are susceptible to noise, misregistration, and source dynamic range due to commonly used decomposition. It is still challenging to generate an activity-level function that relates to all the fusion criteria without lowering algorithm performance. To address this, we use convolutional neural networks (CNN) to create a reliable activity level assessment and weight map calculation.

Over the past decades, research based on deep learning (DL) has also gained attention in the field of image fusion. Deep networks have been employed to address several fusion challenges, including multi-exposure fusion [11], multi-focus fusion [12] and visible/infrared fusion [13]. DL-based approaches can utilize many network branches to achieve differentiated feature extraction and obtain more targeted characteristics. This model can learn feature representations directly from data, allowing for end-to-end fusion process optimization, while eliminating the need for manual feature design.

In deep learning models for image fusion, active level measurements and fusion operations are often performed jointly through architectures like CNN as shown in Fig 2. CNNs analyze images at multiple scales to capture intricate details. Later Fusion Operations combines information from multiple input images to generate a single, fused image. This can involve techniques like weighted averaging, attention mechanisms, or feature concatenation. In these models, CNN learns to weigh the importance of features from different input images and combines them effectively during the fusion process. This joint learning enables the model to adaptively determine the relevance of features and optimize the fusion process accordingly.

The DL based fusion approach captures complex interactions among the data, that lead to superior fusion results in terms of spatial and spectral fidelity, contrast improvement, artifact suppression, and noise reduction. Deep learning-based image fusion algorithms offer notable benefits in terms of performance, adaptability, scalability, and robustness when compared to traditional fusion methods. Furthermore, for the purpose of achieving better performance in image fusion tasks, studies focused on hybrid deep learning models are required since they combine the advantages of different deep learning architecture types [14].

Here we proposed ensemble model with the use of VGG-19 and SNN architecture that extract the most significant features and fuse effectively to result in signal compound image.

## 1.1. The key offerings of our proposed DL based fusion model as follows

- This ensemble model generates fused images with improved contrast, high resolution, and detailed edge information with minimum artefacts for various combinations of image modality.

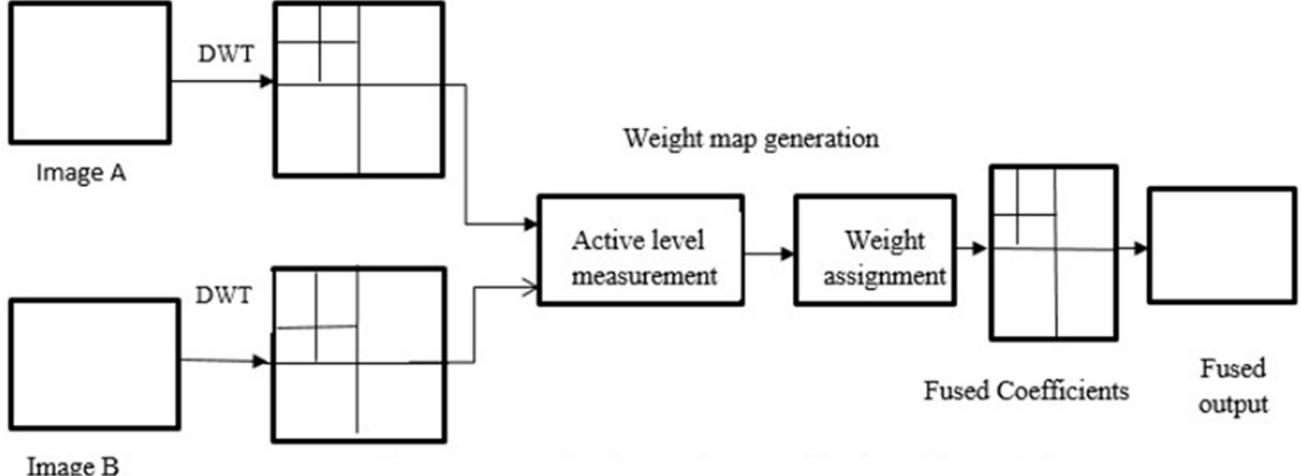

**Fig 1. Conventional DWT decomposition-based image fusion.**

- The proposed model was more robust, and the performance of the proposed model was evaluated for various combinations of source images.

- For all the combinations the model results fused output with improved metrics which are comparatively higher than the existing fusion algorithms.

The paper is organized as follows. Section 2 describes the literature review. Section 3 discusses the proposed DL framework for medical image fusion. The simulated results for various medical image fusions are presented in section 4. Discussion and the conclusion are given in Sections 5 and 6 respectively.

## 2. Literature survey

There are numerous approaches for performing image fusion, which include methods based on spatial domain, frequency domain and deep learning. Bhateja, Vikrant, Abhinav Krishn, and Akanksha Sahu [15], presented a hybrid image fusion model with principal component analysis (PCA) and DTCWT for the fusion of CT-MRI images. The proposed model is designed to improve directionality, shift invariance, and preserve spectral content through DTCWT and improved resolution along with reduced redundancy is attained using PCA. The feature extraction and representation were improved by combining the PCA-DTCWT algorithms and visually high-quality fused output was obtained.

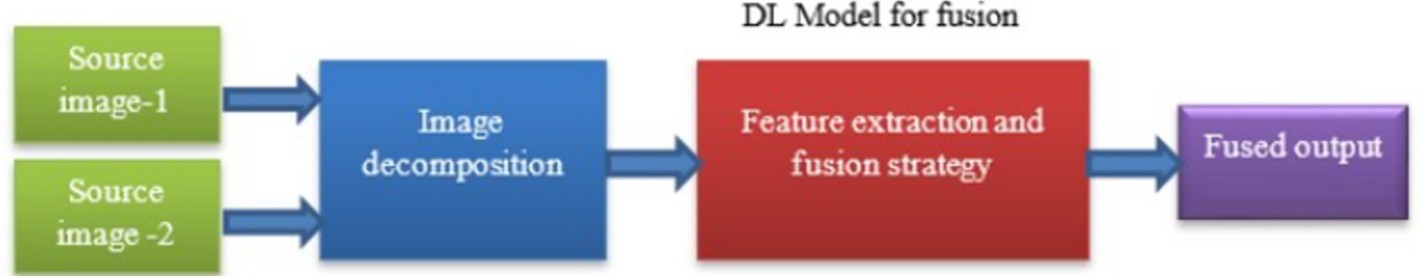

**Fig 2. Conventional DL based image fusion.**

Mehta, Nancy, and Sumit Budhiraja [16], proposed a multimodality image fusion model with a guided filter in the domain of non-subsampled contour let transform (NSCT). In the initial stage, the input images are decomposed into high and low-frequency coefficients. The high-frequency coefficients of various source images are combined using a Guided filter to preserve edge information. Phase congruency was used to combine low-frequency coefficients. The proposed fusion model improved the quantitative metrics in terms of the structural similarity index, and entropy metric.

Liu, Xingbin, Wenbo Mei, et al [17], proposed a multi-modality medical image fusion method based on an image decomposition framework and a non-subsampled shear-let transform. Here, the texture and approximation components are separated using a moving frame-based decomposition framework (MFDF) and a non-subsampled shear let transformation (NSST). Maximum selection fusion rules are used to fuse texture components, approximate components are combined with the use of NSST. Finally, fused output is generated with the components synthesis process. Wen, Yu, et al [18], employed a two-channel CNN-based multifocal image fusion model to derive a clarity map of input images. Using morphological filtering, the clarity map is smoothed and finally, a fused image is constructed by merging the smoothed parts of the input images. The approach succeeded in providing good-quality and quantitative metrics for fused images.

Habeeb and Nada Jasim [19], presented a performance-enhanced model for the fusion of multimodality medical images based on a sharpening Wiener filter (SWF) in combination with discrete wavelet transform (DWT). The model is proposed to overcome the major drawback of fusion alone with DWT, which suffers from blur due to the limited directionality of wavelets. With the use of SWF and DWT, the model enhanced the fusion performance. Joshi et al [20], proposed a medical image fusion (MIF) method with the use of the Stationary Wavelet Transform (SWT) and bidirectional exact pattern matching algorithm (BEPMA) for speckle noise reduction. Input images are decomposed using SWT followed by active level measurement using average and weighted values. Finally, inverse transform is applied to obtain the fused output, with less speckle noise.

Alseelawi, Nawar, Hussein Tuama Hazim, and Haider TH Salim ALRikabi [21], proposed a conventional hybrid approach model based on DTCWT, and non-subsampled contour let transform (NSCT). In this hybrid model, two algorithms are combined to improve the performance and superiority of the model when compared to some other works. It employed a CNN to generate a weight map for the fusion of source images that have pixel movement information from multimodality images. Finally, the performance of the model is evaluated both qualitatively and quantitatively. Wang, Rui, et al [22], proposed a multimodality chromatic image fusion algorithm to combine colour channels through Geometric algebra- discrete cosine transform (GADCT).

The model effectively integrates GA's features, which signify a multi-vector image. A three-stage process is used to get the final fused output: division into several input image blocks and GA multi-vector form expression, the extension of traditional DCT to GA space, Use of GADCT decomposition to get AC and DC coefficients. The proposed model can provide clear and comprehensive fusion images. A convolutional neural network (CNN)-based method for medical image fusion was presented by Liu, Yu, et al [23]. Using the Siamese convolutional network creates a weight map which combines the pixel activity data from two source images. To be more compatible with human visual perception, the fusion process is carried out by employing image pyramids with multiple scales. A local similarity-based method is employed to modify the fusion mode further adaptively for the decomposed coefficient. The proposed method produces promising results in terms of both visual quality and objective assessment.

Lahoud, Fayez, and Sabine Sistrunk [24] 2019 proposed VGG-19 a deep neural architecture for the fusion of medical image modality. A unique approach based on deep feature maps taken from a convolutional neural network is used to merge the images. The multi-modal image fusion technique works by fusion weights that are generated through a comparison of these feature maps. The model has been evaluated with various kinds of source image combinations, not just the fusion of two images. The primary constraints of this suggested model are the fused imaging quality of the source images and the final performance metric value.

Rashmi Dhaundiyal et al. 2020. suggested a multimodality medical image fusion based on clustering. To fuse the approximation layers (coarse layer) and detail layers after the source images have been disintegrated by the Stationary Wavelet Transform method, the Fuzzy Local Information C-means Clustering (FLICM) and Local Contrast Fusion methods are utilized. The approach is demonstrated by both objective and subjective assessments that it retains more specific information in the original images and enhances the quality features and edge preservation of the resultant fused image [25].

A new fusion method for multi-modality medical images was suggested by Diwakar, Manoj, et al. in 2022 to provide an efficient fused image. were low-frequency and high-frequency elements of the medical images obtained using NSCT. Furthermore, a fusion technique based on clustering was employed to fuse low-frequency components with cluster analysis of features. In a similar vein, contrast-preserving image fusion was carried out by directed contrast based on cluster-based components to fuse the high-frequency coefficients. subsequently, a comparison analysis and experimental findings are performed using the multi-modal medical image dataset. The suggested method demonstrates superiority over the state-of-the-art fusion procedures in terms of edge preservation and contrast [26].

Huo, Xiangzuo, et al. proposed an effective three-branch hierarchical multi-scale feature fusion network structure (HI Fuse) for the classification of medical images in 2024. The suggested HI Fuse uses a parallel hierarchy of local and global feature blocks with linear computational complexity related to image size and flexibility to model at different semantic scales, effectively capturing local features and global representations. Furthermore, an adaptive hierarchical feature fusion block (HFF block) is developed to fully utilize the features that have been acquired at various levels in the hierarchy. The HI Fuse model performs better than other comprehensive models. 9.4% on the Kvasir dataset, 21.5% on the Covid-19 dataset, and 7.6% on the ISIC2018 dataset surpasses the baseline [27].

A new multi-modal medical image fusion framework utilizing co-occurrence filters and local extrema in the NSST domain was presented by Diwakar, Manoj, Abhishek Singh, and Achyut Shankar in 2021. The suggested approach employs the Non-subsampled Shear Let Transform (NSST) to decompose input images into low- and high-frequency components. A Co-occurrence filter (CoF) in low-frequency components is utilized to fuse the base layers and detail layers employing a distinctive local extrema (LE) technique. Sum Modified Laplacian (SML) is used in high-frequency components as an edge-preserving image fusion approach to fusing the high-frequency coefficients. The suggested approach performs better than cutting-edge fusion procedures when taking edge preservation into account in both subjective and objective evaluations [28].

Diwakar, Manoj, et al. 2023 presented a multimodal image fusion method using non-subsampled shear let transform (NSST) and modified sum-modified Laplacian (MSML). The feature high and low-frequency image components are extracted initially with NSST later this frequency component are fused using MSML. The proposed model holds detailed edge information with improved imaging quality. Overall, the proposed model results almost 10% better than existing approaches in terms of standard deviation and mutual information along with

improved visual quality in terms of texture preservation, edge preservation and more information [29].

A review of non-conventional methods of multi-modality-based image fusion was demonstrated by Diwakar, Manoj, et al in 2023. This paper presents a critical evaluation of multi-modality-based image fusion, highlighting some significant non-conventional work, as well as a discussion of the benefits and drawbacks of multi-modality image fusion. The paper also covers some feature directions in multi-modality image fusion from conventional to some deep learning techniques. The paper focuses on various multi-modality image fusion methods based on transformed domain and spatial domain. Finally, the paper ends up proving that a transformed domain strategy produces better results for analogue spatial domain schemes in terms of visual effects, the performance measurements [30].

In 2022 Diwakar, Manoj, et al. presented a new image fusion technique in the shearlet domain to improve existing methods for the internet of medical things. In this work, NSST coefficients for low and high frequencies are first generated from both input images. Later the detailed features of frequency components are obtained by performing a new local extremum (MLE) based decomposition. A saliency-based weighted average is applied to these MLE features with a co-occurrence filter to enhance low frequency NSST coefficients, then these Coefficients are fused with a weighted fusion strategy. The local type-2 fuzzy entropy-based fusion is performed to fuse high frequency NSST Coefficients. At last inverse NSST is applied to get the resultant fused output with improved visual and performance metrics [31].

Khan, Sajid Ullah, et al. proposed a similar work in 2023 on multimodal medical picture fusion in the NSST domain with structural and spectral feature improvement. Initially, two pairs of images are created using the intensity, hue, and saturation (IHS) approach. A Non-Subsampled Shearlet Transform (NSST) is used to decompose the input image to corresponding low and high frequency components. Next, a suggested Structural Information (SI) fusion approach and Principal Component Analysis (PCA) are utilized for applying a fusion rule to low and high-frequency components. Finally, inverse NSST and HIS are performed to get fused output, and the proposed model was validated using different modalities containing 120 image pairs to show the superiority of the model both in qualitative and quantitative performance [32].

The major limitations in the existing works are as follows 1) Image fusion with conventional methods is usually not robust due to several factors, such as generating noise and mis-registration. 2) Complex mapping from the source image to the weight map is the most complicated process in the many existing methods. 3) fused images with improved image quality in terms of contrast, resolution, hold of edge information along with improved metrics is always a crucial task. To overcome the above limitation, we proposed a DL-based image fusion model that effectively generates fused output with good imaging quality. The proposed DL model with the use of SNN in combination with VGG-19 can effectively combine the base and detailed features automatically to generate a weight map (W) to improve fusion performance.

## 3. Proposed dl framework for image fusion

The main challenge associated with conventional algorithms is their inability to build the two critical stages of fusion processes and to extract significant features with minimum artefacts. In this paper, we proposed a deep neural framework with SNN and VGG-19 to fuse multimodality images automatically with improved imaging quality. This hybrid architecture of the pre-trained deep CNN model (VGG-19) and non-pretrained CNN (SNN) is implemented using the stacking ensemble method that leverages the advantages of both architectures. The model can capture specific features learned by the non-pretrained model and general features learned

by the pretrained, model significantly hold the more useful information. In a deep learning model, particularly in the context of active-level measurements and fusion, the process typically involves integrating multiple sources of information into a single unified framework. This integration can be achieved using multi-input neural networks, attention mechanisms, or recurrent neural networks. Here the extracted features are fused using a weighted sum and multi-layer fusion strategy to generate a final fused single compound image with improved imaging quality. Further, the performance of the designed framework is evaluated with various existing fusion models for different combinations of source images. Fig 3 depicts the proposed DL-based image fusion framework.

It is observed that the model contains three main steps: image decomposition, auto-weight map calculation, and image reconstruction. The process of fusing various image modalities is done using the steps mentioned below

Step-1: Image registration is used to align input images to have the same coordinates to avoid miss alignment losses.

Step 2: The aligned source images are decomposed using the DTCWT algorithm to produce base and detailed content.

Step-3: Base features of both sources are given as input to the trained Siamese network to extract and concatenate the base features of both sources. The more detailed features are extracted with the use of a pretrained VGG-19 model.

Step-4: The extracted output features of the SNN and VGG-19 model are combined and inverse DTCWT is applied to generate the final fused output.

Step-5: For the fused output the performance metrics were evaluated and compared with the existing method.

The proposed DL-based fusion framework performs well in the accurate fusion of images and effectively holds the crucial information from images with minimal undesirable visual distortions. The workflow of the proposed DL-based fusion models is depicted in Fig 4, and sections 3.1 to 3.7 provide a detailed description of the proposed fusion model starting from acquiring source image, pre-processing of source image, auto weight map generation using

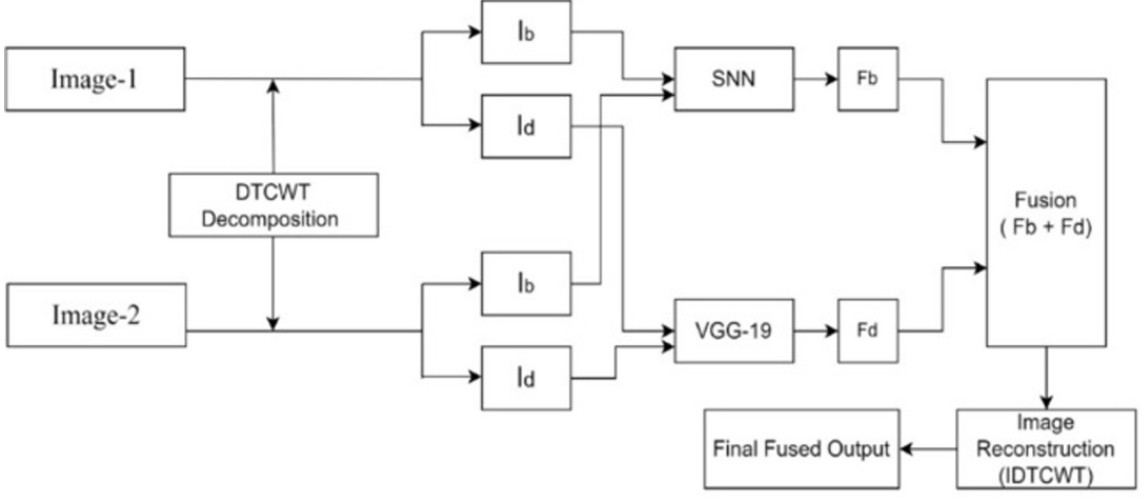

**Fig 3. Proposed model for image fusion.**

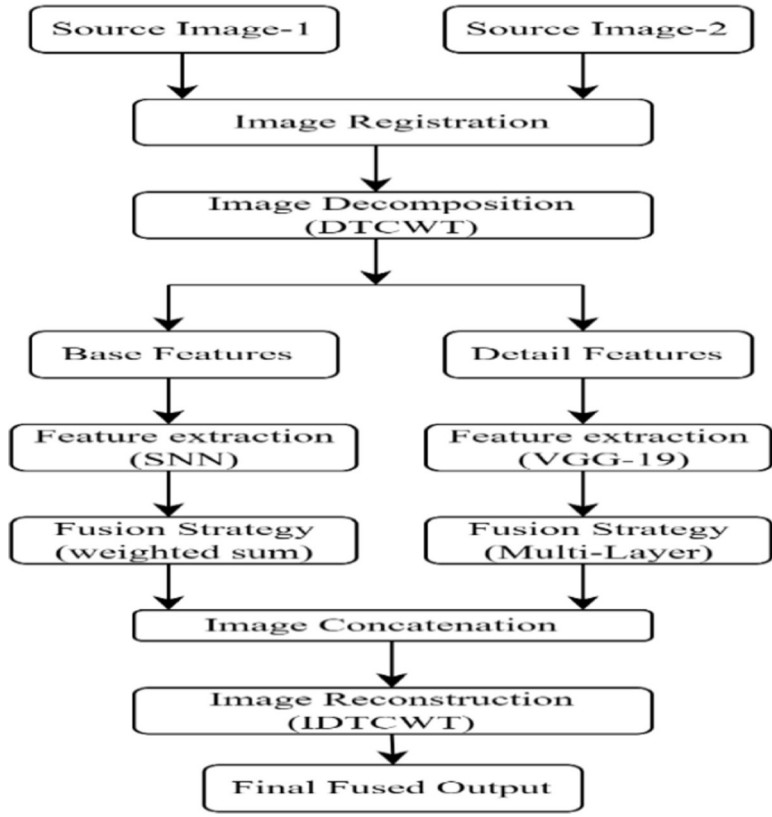

**Fig 4. Flow of the proposed model.**

SNN and VGG-19 feature extraction model, image reconstruction followed by performance evaluation.

## 3.1 Dataset

In the proposed model we conducted multiple experiments with the most recent, publicly available standard data sets. For fusion of various image modality, we have considered the data set available in given mentioned link.

https://github.com/praneethMohan/GIST-CT-PET/tree/main, https://github.com/MorvanLi/imagefusion-zoom/tree/main/Medical_Image_Fusion_Methods/Havard-Medical-Image-FusionDatasets, https://www.kaggle.com/datasets/mateuszbuda/lgg-mri-segmentation?rvi=1.

## 3.2 Pre- Processing

Clinical imaging experts employ pre-processing to enhance the appearance of source images and identify the best features. Some of the most crucial pre-processing techniques are noise reduction, image resizing, rotation, scaling, and contrast enhancement to increase the quality and quantity of the datasets. The aim of pre-processing is an improvement of the image data that suppresses unwilling distortions or enhances some image features important for further processing.

Here we mask use of preprocessing techniques as shown in Fig 5 to improve the image quality. Image registration aligns the geometrical coordinates of two images and matches their

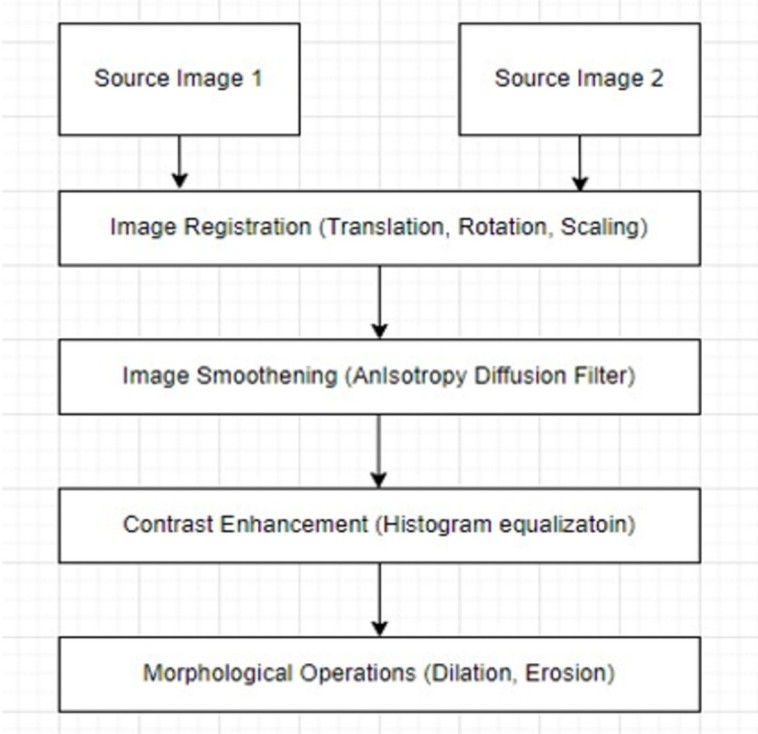

**Fig 5. Preprocessing flowchart for proposed work.**

intensity values with some commonly used techniques like translation, rotation, and scaling to avoid artefacts because of misalignment [33]. An anisotropy diffusion filter is employed to improve the edges in the images [34]. Morphology adds pixels to image boundaries (called dilation) and removes pixels from image boundaries (called erosion) improving image visualization [35]. For better performance, histogram equalization (HE) is used for image enhancement [36].

## 3.3 Image Registration (IR)

The process for determining the geometric transformation connecting identical (anatomical) features in two image series is generally referred to as image registration. It transforms various types of data into a single coordinate system. It involves the use of feature-based registration, intensity-based registration, or a mix of the two to register two or more images to align their associated features spatially. Feature-based methods use key points or landmarks in the images to perform alignment. Pixel-based alignment, also known as intensity-based alignment, involves directly comparing the pixel intensities of two images to achieve alignment. Pixel-based alignment is useful when feature-based methods fail due to a lack of distinct features or when high-accuracy alignment is needed. In pixel-based alignment, the processes of alignment start with a rough initial alignment of the two images and can be accomplished either manually or using a coarse alignment method.

An optimization technique to refine the alignment and cost function that measures the discrepancy between the pixel values of the two images. This optimization process adjusts the transformation parameters (e.g., translation, rotation, scaling) to minimize the misalignment. Generally, we use the sum of squared differences (SSD), and mutual information (MI) as a

cost function to quantify the difference between corresponding pixels in the two images. Iteratively adjusting the transformation parameters is done to minimize the cost function until convergence is achieved. The wrapping method is applied to the final transformation of one of the images to align it with the other and additional refinement techniques such as interpolation to improve the quality of the aligned images. Fig 6 exhibits the output of image registration processes of source images using pixel level alignment with the same size, and resolution and aligned properly to avoid miss alignment loss in the performance of the model.

## 3.4 Decomposition of source images

Image decomposition is the critical initial phase in fusion where the source image must be decomposed into various components using filters before it is fused. Here, for the decomposition of source images, we consider Dual-tree complex wavelet transform (DTCWT). In medical imaging, DTCWT can be used for multiscale decomposition, which involves breaking down an image into different frequency bands to extract useful information at various scales. Applying the DTCWT to the original image involves filtering the image with a set of wavelet filters. The filters that are mostly used in DTCWT is Bilateral filter which was proposed by Tomasi and Mandu chi. It makes use of two Gaussian filters: spatial filtering and range filtering. The spatial filter deals with pixel location while the range filter deals with pixel intensities. Spatial filtering is like the conventional methods, where the neighboring pixels that are present in the vicinity have influence on each other. Spatial filter is the Gaussian filter which can be implemented with different kernel size (say 5x5, 7x7, 11x11). Theoretically Gaussian distribution is non-zero for an infinite range and will require a kernel of infinite length for convolution. In practice as the kernel length increases, the standard deviation becomes negligible. Hence, the effective kernel size is chosen as per the requirements as 7x7. The range filtering is non-linear. The range filter assigns higher weightage to the pixel which has intensity like the center pixel. At the edges, where there is a large variation in the intensity of the pixels, range filtering reduces the influence of the distant pixels thereby preserving the edges.

DTCWT employs two separate trees (real and imaginary) to capture both magnitude and phase information. Two sets of filter banks are arranged in rows and columns in the upper tree of the DTCWT. They are designated as h0(n) and h1(n), respectively, denoting the low pass and high pass filters. Like this, the lower tree features unique high pass g1(n) and low pass g0(n) filters. The sub bands of each decomposition level are then provided by each pair of low pass and high pass filters. The wavelet associated with the upper tree is $\varphi_u(t)$ is real part of complex wavelet and the wavelet associated with the lower tree is $\varphi_1(t)$ is the imaginary part of the complex wavelet. The complex wavelet is illustrated as $\varphi u(t)+j\varphi l(t)$, where $\varphi u(t) = HT(\varphi l(t))$. This decomposition allows for enhanced analysis, feature extraction, denoising, and compression of medical images.

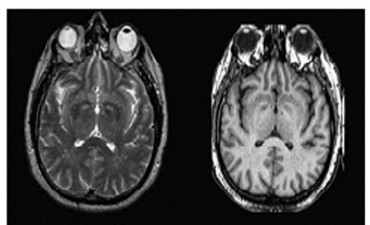 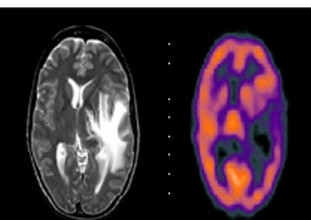 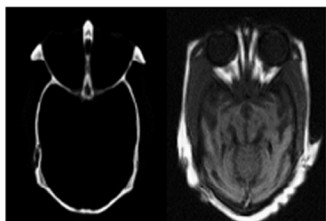 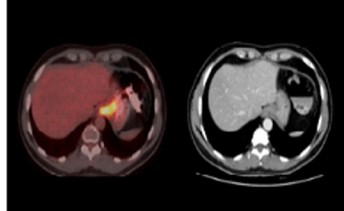

**Fig 6. Four pairs of register source images used in experiment.**

## 3.5 Proposed fusion work

The fusion processing of base parts and detail content is introduced in the next subsections. In our work, we use K = 2 for preregistered source images, the fusion methodology applies to K > 2. The source images will be referred to as Ik, with k ranging from 1 to 2. The proposed DTCWT outperforms other image decomposition methods such as wavelet decomposition and latent low-rank decomposition which is more effective and can save time. The source image Ik is divided into base components (Ibk) and detail content (Idk), separated by DTCWT-based decomposition. The base components are obtained by solving a particular optimization problem:

$$I_K^b = arg \min_{I_K^b} ||I_k - I_K^b||_F^2 + \lambda(||g_x * I_K^b||_F^2 + ||g_y * I_K^b * I_K^b||_F^2) \tag{1}$$

where $g_x = [-1\ 1]$ and $g_y = [-1\ 1]^T$ are the horizontal and vertical gradient operators, respectively. After we get the base parts $I_K^b$, the detail content is obtained by Eq 2

$$I_K^d = I - I_K^b \tag{2}$$

As shown in Fig 3, the source images are denoted as I1 and I2. Firstly, the base part and the detail content for each source image are obtained by solving Eqs (1) and (2), where k ∈ {1, 2}. Following the weighted averaging strategy's fusion of the base elements, the vgg-19 deep learning framework reconstructs the detailed content. Lastly, the fused base component Fb and detail content Fd will be added to reconstruct the fused image F.

**3.5.1 Fusion of base part using SNNs.** It is widely accepted that CNNs are far superior to conventional methods in a wide range of visual recognition problems because CNNs can learn the most useful features from a large amount of training data [37]. CNNs can successfully carry out image fusion operations integrating a variety of image modalities to improve the quality of medical images. Based on activity level measurement and weight assignment concepts, several CNN-based architectures for image fusion approaches have recently been presented [38]. SNN is one of the most prominent or similar groups of neural network architectures with two or more related sub-networks placed parallel to each other. Here similar means that the networks have the same configuration, same parameters, and weights. Generally, these types of network architectures are used to find the similarity among the inputs by comparing their feature vectors [39].

Due to their unique properties, SNNs are trained to extract the feature vector from different source images that are given as input to the identical sub-networks in the SNN. The extracted features from the bi-lateral structure for source images are concatenated to get base information of source images. Fig 7 depicts the bilateral architecture that was designed for extracting and combining the base information for various source images. The structure was designed with the use of a convolution layer, max pooling layer, fully connected layer, drop-out layer, and local response normalization with ReLU activation. Table 1 provides the entire architecture that includes different kernel sizes.

Selecting an image patch size is a very crucial process. In general, there's a trade-off between patch size and performance. Since large image features are encoded by the neural network with a greater patch size, the accuracy is higher, but the efficiency is negatively impacted. The training accuracy using a small patch size is not robust. Considering the constraints and the size of the dataset, we used 16 × 16 patches in our proposed work. The obtained 256 feature maps from each branch were concatenated into one 256-dimensional feature vector after being fully connected. Then, for SoftMax operation, 2-dimensional vectors are further

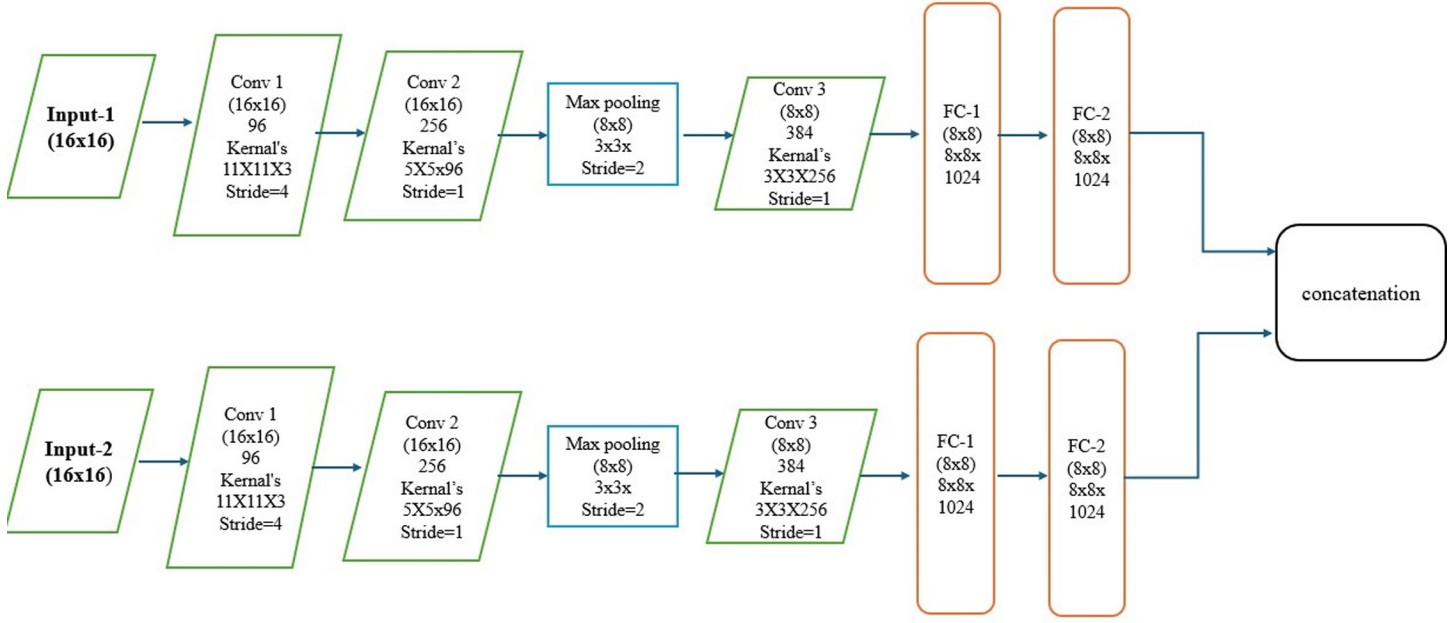

**Fig 7. Siamese network architecture for base part fusion.**

completely connected with the initial fully connected layer to generate the probability score of two classes.

**3.5.2 Weight map generation (W) for base part.** Calculating a weigh map of pixel activity across multiple source images is one of the primary obstacles in image fusion. In this model, we consider various medical images for image fusion tasks. We intend to generate a weight map for base information and the resultant value should range from 0 to 1 for training the SNN model. The weight map coefficients that are produced are considered as the fusion rule that specifies the percentage of each pixel intensity value in the source image that corresponds to that pixel in the weighted sum step. The input image pairings are encoded by a Siamese network and a score is given to each pair that reflects the saliency of each source. The probability calculated through the SoftMax operation eventually becomes a weight value in the weight map. Using the CNN model, a weight map similar in size to the input image pair is generated.

The weighted sum pixel calculation equation in fusion processes is significant because it allows for the integration of information from multiple input images or data sources. The

**Table 1. Detailed SNN parameter settings.**

| Sl. No. | Layer | Size | Kernel size | stride | Padding |
|---|---|---|---|---|---|
| 1 | Input layer | 16x16 | - | - | - |
| 2 | Conv 1 | 16x16 | 11x11 | 1 | - |
| 3 | Conv 2 | 16x16 | 5x5 | 1 | 2 |
| 4 | Max pooling | 8x8 | 3x3 | 2 | - |
| 5 | Conv 3 | 8x8 | 3x3 | 1 | 1 |
| 6 | Concatenation | 8x8 | N/A | - | - |
| 7 | Fully connected 1 | 8x8 | 8x8 | - | - |
| 8 | Fully connected 2 | 8x8 | 8x8 | - | - |

bright pixel in a weight map W indicates a value that is near 1, while the darker pixel shows a value closer to 0. The weighted sum pixel value assigned the weights to every pixel in the source images, which produces a resultant image with a weighted sum of every pixel value in source images as given in Eq 3. Fb (x, y) is the weighted sum of source images I1 and I2 at coordinates of (x, y) respectively.

$$Fb (x, y) = \alpha1 \, I_1^{\,b} (x, y) + \alpha2 \, I_2^{\,b} (x, y) \tag{3}$$

where, (x, y) = Corresponding position of the image intensity in source image.

$\alpha1$ & $\alpha2$ = Represents weighing factor for respective source images.

The weighting process is crucial for achieving optimal fusion results that reflect the desired characteristics from the input images. It determines the contribution of each source image to the fused base part in image fusion. By adjusting the weighting factors, image fusion algorithms can control how much each source image contributes to the fused base part, allowing for the preservation of important information while minimizing artefacts or inconsistencies.

If a source image is assigned a high weight, it means that its information is considered more important or reliable for the fused base part. As a result, this source image will contribute more significantly to the final fused base part, potentially dominating its characteristics. If Source images with low weights contribute less to the fused base part compared to those with higher weights. Their influence on the result is minimized, and they may only marginally affect certain regions or features. When a source image is assigned a moderate weight, its contribution to the fused base part is balanced with other sources. While it still influences the result, its impact may not be as pronounced as that of sources with higher weights. Here we consider $\alpha1$ & $\alpha2$ as 0.5 each to preserve the common features from both the source images and reduce the redundant information.

**3.5.3 Fusion of detailed content with VGG-19.**  In our proposed fusion work, we used CNN with extreme deep layers (VGG-19 up to 19 layers) for to extract detail content $I_1^d$ and $I_2^d$. This procedure is shown in Fig 8. It is noted that VGG-19, which is 19-layer architecture implies that the model has 19 weights that can be learned to generate feature maps. From the figure, it is very clear that we used VGG-19 to extract multi-layer deep features from the source

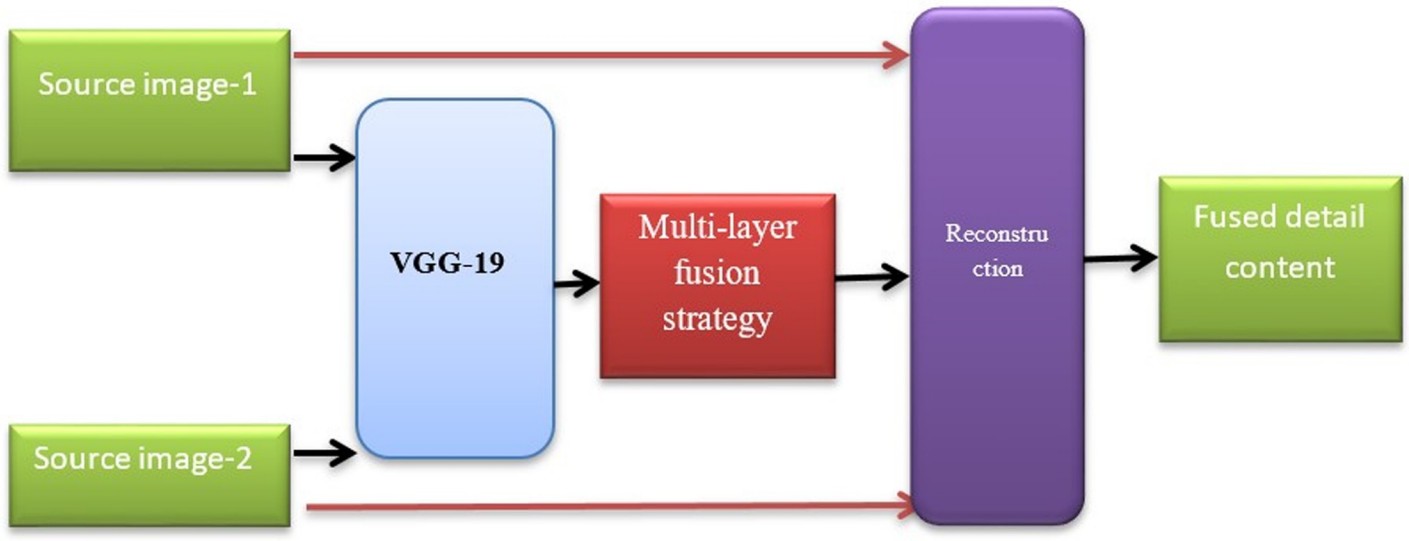

**Fig 8. Procedure for detailed content fusion with VGG-19.**

image. We obtained a weight map of source images by a multi-layer fusion strategy. The detail content is reconstructed by obtaining weigh map and detail content on pre-trained VGG-19 network.

The multi-layer fusion strategy for detailed features is explained here. Let detail content $I_k^d$. $\phi_k^{i,n}$ indicates the feature maps of k-th detail content extracted by the i-th layer and n is the channel number of the i-th layer, $n \in \{1, 2, \cdots, N\}$, $N = 64 \times 2i-1$,

$$\phi_k^{i,n} = \phi_i(I_k^d) \tag{4}$$

Each $\Phi i(\cdot)$ denotes a layer in the VGG-19 network and $i \in \{1, 2, 3, 4\}$ represents the Relu 1–1, Relu 2–1, relu 3–1 and relu 4–1, respectively. Let $\phi_k^{i,1:N}$ k (x, y) denote the contents of $\phi_k^{i,n}$ at the position (x, y) in the feature maps. As we can see, $\phi_k^{i,1:N}(x, y)$ is an N -dimensional vector.

After getting deep features $\phi_k^{i,n}$, the activity level map $C_k^i$ will be calculated by L1-norm and block-based average operator, where $k \in \{1, 2\}$ and $i \in \{1, 2, 3, 4\}$. The L1-norm $\phi_k^{i,1:N}(x, y)$ will be the activity level measure of the source detail content. Thus, the initial activity level map $C_k^i$ is given by 5.

$$C_k^i(x, y) = ||\phi_k^{i,1:N}(x_1 y)||_1 \tag{5}$$

The block-averaging operator are used to get the final activity level map $C_k^i$ to make our proposed fusion processes robust to misregistration.

$$\hat{c}_k^i(x, y) = \frac{\sum_{\beta=-r}^{r} \sum_{\theta=-r}^{r} C_k^i(x + \beta, y + \theta)}{(2r + 1)^2} \tag{6}$$

where r-determines the block size. If the r is greater, the fusion approach will be less vulnerable to misregistration, but some detail might get lost. Therefore, r = 1 in our strategy.

Once after obtaining the activity level map $\hat{c}_k^i$, the initial weight maps $w_k^i$ will be calculated by soft-max operator, as given by Eq 7. K represents the number of activity level map, which in our work is set to K = 2. $w_k^i(x, y)$ indicates the initial weight map value that ranges from [0,1].

$$w_k^i(x, y) = \frac{\hat{C}_k^i(x, y)}{\sum_{n=1}^{k} \hat{C}_n^i(x, y)} \tag{7}$$

In the propose VGG-network, the stride of the pooling operator is 2. Therefore, in different layers, the size of feature maps is $1/2^{i-1}$ times the detail content size, where $i \in \{1, 2, 3, 4\}$ indicates the layers of relu 1–1, relu 2–1, relu 3–1 and relu 4–1, respectively. We employ an up-sampling operator to adjust the weight map size to the input detail content size once we obtain each initial weight map, $w_k^i$ k. The up-sampling operator will yield the final weight map, $\hat{w}_k^i$ whose size is equal to the input detail content size. The final weight map is calculated using Eq 8.

$$\hat{w}_k^i(x + p, y + q) = w_k^i(x, y) \tag{8}$$

$$p, q \in \{0, 1, \cdots (2^{i-1} - 1)\}$$

Now we have four pairs of weight maps $\hat{w}_k^i$, $k \in \{1, 2\}$ and $i \in \{1, 2, 3, 4\}$. For each pair $\hat{w}_k^i$, the initial fused detail content is obtained by Eq 9. The initial fused detail content is generated for each pair of weight maps and the input detail content.

$$F_d^i(x, y) = \sum_{n=1}^{k} \hat{w}_n^i(x, y) \times I_n^d(x, y), K = 2 \tag{9}$$

To produce the final fused details content $F_d$ symbolized in Eq 10, in which we choose the maximum value from the four initial fused detail contents for every angle.

$$F_d(x, y) = max\,[F_d^i(x, y)|i \in \{1, 2\}] \tag{10}$$

### 3.6 Image reconstruction

The final stage of the proposed model is the fusion of significant features derived from the SNN and VGG-19 architecture to obtain the fused image. Once we have the fused detail content Fd from the VGG-19, we will combine it with the obtained fused base part Fb from SNN to reconstruct the final fused output for the given source image, wherein expression 11 is given.

$$F(x, y) = F_b(x, y) + F_d(x, y) \tag{11}$$

Eq 11 renders it extremely evident that the final fused output produced is the sum of the fused outputs of the base part and detail content from the proposed deep learning model. Finally, an inverse DTCWT is applied to resultant fused image. It is observed that the proposed model generates a weight map with minimum artefacts results in high quality of fused output.

## 4. Experimental results

### 4.1 Performance metrics

To assess the efficacy of the suggested model, fusion measures such as mean square error (MSE), peak signal noise ratio (PSNR), normalized mutual information *(Q_MI)*, structural similarity index measurement (SSIM), and multi-scale scheme (Q_M) are frequently used [2].

**4.1.1 Peak Signal to Noise Ratio (PSNR).**   PSNR is one of the most important evaluation metrics considered for image fusion. The term PSNR refers to the ratio of the strongest possible signal-to-noise, which greatly determines the quality of image representation. Determining the MSE is crucial for the computation of PSNR and a higher PSNR value indicates greater image quality. It is mainly used to estimate the quality of an image; the maximum value indicates a noise-free image. The PSNR value can be obtained from (12).

$$PSNR = 20 \log_{10} \frac{max_i}{\sqrt{MSE}} \tag{12}$$

**4.1.2. Structural Similarity Index Measurement (SSIM).**   The metric used for assessing the extent to which two images resemble one another is known as SSIM. It is a very important metric in performance evaluation in image fusion. The SSIM value always ranges from -1 to 1, with 1 denoting a perfect match and -1 denoting a total mismatch. The mathematical representation for SSIM is given in Eq 13.

$$SSIM(x, y) = \frac{(2\mu_x\mu_y + 1)(2\sigma_{xy} + C_2)}{(\mu_x^2 + \mu_y^2 + C_1)(\sigma_x^2 + \sigma_y^2 + C_2)} \tag{13}$$

**4.1.3. The Normalized Mutual Information (Q_MI).**   To determine how well the source and fused images' image intensities line up, the Normalized Mutual Information is a crucial fusion metric. A greater Q_MI value indicates higher quality of image.

$$M_F{}^{AB} = MI\,(A,\ F) + MI\,(B, F) \tag{14}$$

**4.1.4. Mean Square Error (MSE).**   Another metric used to assess the quality of the fused image is the mean square error. It represents the sum of the squared errors between the fusion

and source images. The model functions better the lower its value.

$$\left(\frac{1}{N}\right) * \sum ((I[i,j] - K[i,j])^2) \tag{15}$$

N is the total number of pixels in the image

**4.1.5. The multi-scale scheme (Q_M).** The edge data is located and extracted by using the band-pass and high pass components of the decomposition.

## 4.2 Experimental findings

PSNR is one of the most important evaluation metrics considered for image fusion. The term PSNR refers to the ratio of the strongest possible. This section presents findings from experiments, that serve as evidence of the effectiveness of the suggested DL model for the fusion of diverse image modalities. Here, we combined the four pairs of source images obtained from various imaging techniques, including CT, MRI, MR-T1, MR-T2, and PET scans. The experiments were carried out using the Python 3 Google Compute Engine backend (GPU) of the Google Colab Platform. Fig 9 and Table 2 show the experimental findings for the proposed DL model for various kinds of input sources.

The model achieved high-quality fused output for the fusion of various source image pairs. For all the combination of the source images the model results in high visual quality improved interpretation, resolution with clear edge information. The fused image showed improved results for measures, including PSNR, SSIM, Q_MI, Q_M, and low MSE value, for all combinations.

## 4.3 Comparison with other state-of-art techniques

To prove the superiority of the proposed deep learning-based image fusion model we performed a state-of-art comparison of various image fusion methods for various combinations of source images. The proposed DL-based fusion model with SNN and VGG-19 outperformed well in terms of both qualitatively and quantitatively compared to other fusion algorithms such as Discrete Wavelet Transform (DWT), Non-Subsampled Contour let Transform (NSCT), Dual-Tree Complex Wavelet Transform (DTCWT)+ NSCT, GADCT, PCA

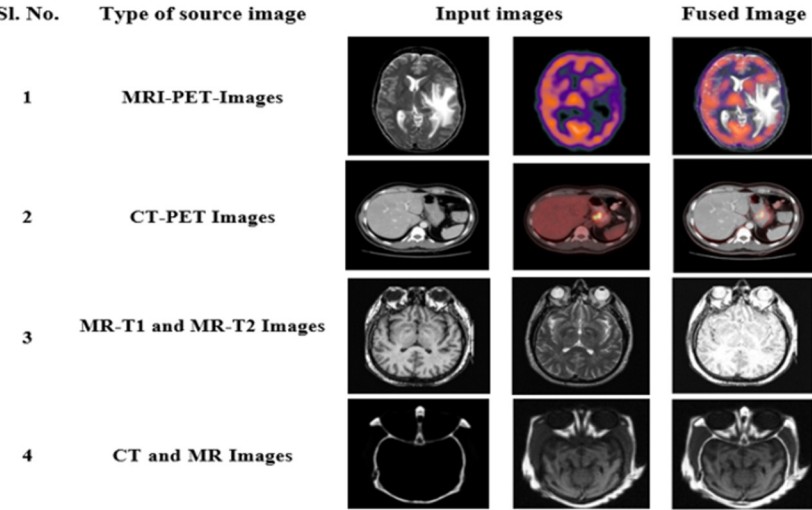

**Fig 9. Resultant fused images for various combinations of source images.**

**Table 2. Performance metrics for various combinations of source images.**

| Sl.no | Source Combinations | SSIM | MSE | PSNR | Q_MI | Q_M |
|-------|--------------------|------|-----|------|------|-----|
| 1 | MRI-PET-Images | 0.927 | 28.55 | 34.54 | 0.864 | 0.80 |
| 2 | CT-PET Images | 0.933 | 33.02 | 35.44 | 0.820 | 0.84 |
| 3 | MR-T1 and MR-T2 Images | 0.927 | 29.90 | 34.36 | 0.844 | 0.86 |
| 4 | CT and MRI Images | 0.947 | 31.82 | 32.95 | 0.852 | 0.79 |

+ DTCWT, along with few pre-trained models like conventional neural networks (CNN) and Visual Geometry Group (VGG-19).

**4.3.1 Fusion of MRI and PET images.** We considered the brain MRI and PET images to carried out image fusion operation to demonstrate the suggested DL's effectiveness in comparison to another fusion method. Fig 10 and Table 3 Shows the resultant visual and objective metrics for the fusion of MRI and PET images and Fig 11 replicates the comparative analysis graph offusion metrics such asMSE, Q_MI, Q_M, PSNR,and SSIM metrics concerning MRI and PET images. The performance metrics have been compared to those of existing fusion algorithms.

**4.3.2 Fusion of CT and PET images.** Here we consider a pair of CT and PET images of gastrointestinal stomal tumors (GISTs). The simulated results proved that the model performs well in terms of qualitative and quantitative features with high PSNR, SSIM, Q_MI and Q_M values of 35.44, 0.93, 0.820 and 0.84 which is comparatively higher than other models. Also, it produces an MSE value of about 33.02 that is much lower than previous fusion algorithms, which implies the fused image produced having less artifacts and superior image quality as shown in Fig 12 and Table 4. Fig 13 analysis of comparison graph demonstrates how effectively the model outperformed well compared to existing fusion algorithms.

**4.3.3 Fusion of MR-T1 and MR-T2 images.** Here for fusion purposes, we consider MR-T1 and MR-T2 source images from MRI database. Fig 14 and Table 5 replicate the visual and quantitative representation of the fusion of MR-T1 and MR-T2 using various fusion

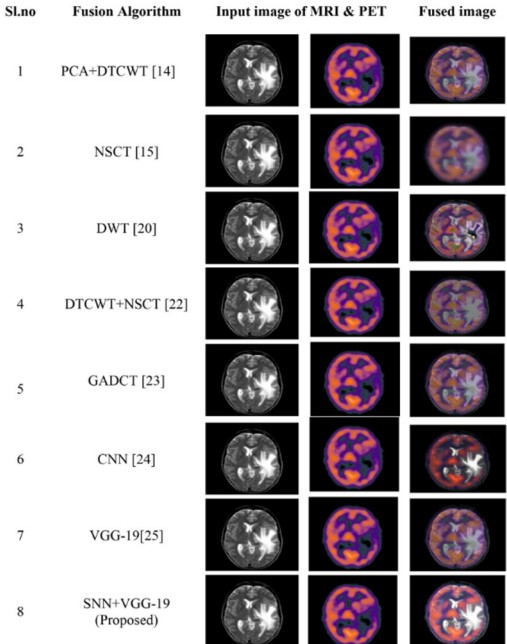

**Fig 10. Comparative analysis of proposed model with various existing fusion models for fused MRI & PET image.**

**Table 3. Resultant objective metrics of fused MRI-PET of proposed model in comparison with for various existing fusion algorithms.**

| Fusion Algorithms | PERFORMANCE METRICS | | | | |
|---|---|---|---|---|---|
| | SSIM | MSE | PSNR | Q_MI | Q_M |
| PCA+DTCWT [13] | 0.772 | 49.86 | 31.89 | 0.683 | 0.560 |
| NSCT [14] | 0.706 | 68.77 | 31.33 | 0.664 | 0.672 |
| DWT [19] | 0.763 | 58.34 | 31.41 | 0.723 | 0.691 |
| DTCWT+NSCT [21] | 0.776 | 42.34 | 31.41 | 0.812 | 0.744 |
| GADCT [22] | 0.814 | 44.56 | 32.75 | 0.731 | 0.783 |
| CNN [23] | 0.792 | 38.15 | 32.09 | 0.786 | 0.621 |
| VGG-19 [24] | 0.813 | 32.42 | 32.98 | 0.798 | 0.763 |
| SNN+VGG-19 (Proposed) | **0.927** | **28.55** | **34.54** | **0.864** | **0.882** |

techniques and compare them to the proposed model. Fig 15 is a comparison analysis graph illustrating the proposed model's performance concerning fusion metrics such as PSNR, SSIM, Q_MI, Q_M, and MSE. The model results in high PSNR value of 34.54, SSIM- 0.927, Q_MI- 0.844, Q_M-0.86 and low MSE-29.20. According to the quality measures, it has been concluded that the proposed algorithm can fuse specific MR-T1 and MR-T2 images with good imaging quality.

**4.3.4 Fusion of CT and MR images.** In this instance, we combine CT and MRI images to perform fusion operation. The experimental results demonstrated that the proposed model has improved performance metrics, which are significantly higher than those of existing fusion approaches as shown in Table 6. The proposed model also produces high-quality fused images with good resolution, along with detailed edge information. detailed information as shown in Fig 16. Comparative analysis graphs for objective metrics for CT & MRI image fusion are shown in Fig 17.

## 5. Discussion

Medical images play a crucial role in the characterization and diagnosis of various tumours. Several situations in medical image analysis require a simultaneous view of both

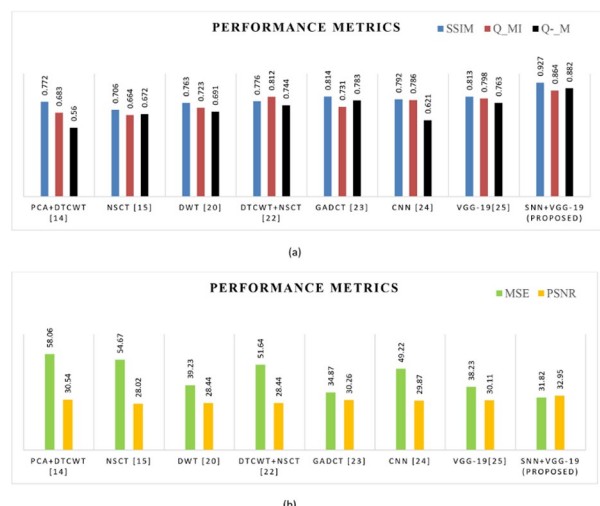

**Fig 11.** (a) & (b). Comparative analysis of objective metrics with other algorithms for the fusion of MRI and PET image.

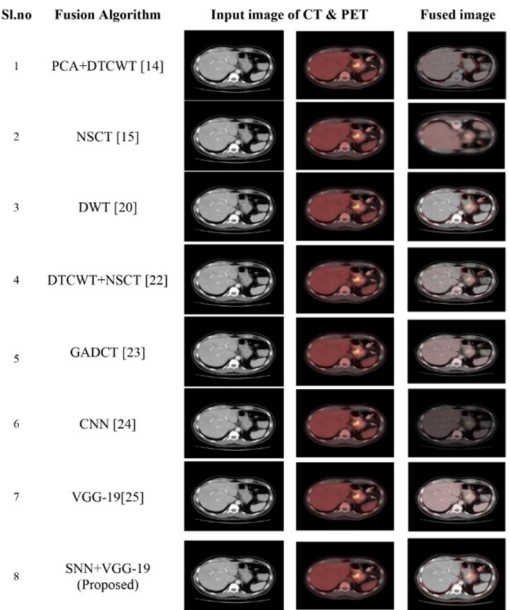

**Fig 12. Comparative analysis of proposed model with various existing fusion models for fusion of CT and PET images.**

structural and functional information in a single image. Auto and complex feature extraction can benefit from a DL-based image fusion model which is less than conventional fusion methods. The proposed ensemble model with the use of SNN in combination with VGG-19 leverages the advantage of both the architecture, making the model more powerful in the effective fusion of various source images. SNN has two identical networks which consist of convolution and 2hidden layers help to extract hierarchical features, and further improve the image quality concerning parameters. On the other hand, the Vgg-19 model is convolutional neural, which is already pre-trained by millions of images from ImageNet. This model consists of a deep neural network, which helps to extract high features. Due to its simplicity and effectiveness, it is best suitable for image classification tasks. As these two networks are familiar with feature extraction, in our proposed model we mask the use of models to produce a fused image quality with improved visual quality and performance metrics.

**Table 4. Resultant objective metrics of fused CT-PET of proposed model in comparison with for various existing fusion algorithms.**

| Fusion Algorithms | PERFORMANCE METRICS | | | | |
|---|---|---|---|---|---|
| | **SSIM** | **MSE** | **PSNR** | **Q_MI** | **Q_M** |
| PCA+DTCWT [13] | 0.791 | 56.23 | 32.68 | 0.642 | 0.661 |
| NSCT [14] | 0.627 | 62.64 | 30.92 | 0.601 | 0.573 |
| DWT [19] | 0.781 | 60.11 | 31.54 | 0.734 | 0.680 |
| DTCWT+NSCT [21] | 0.743 | 51.55 | 31.02 | 0.772 | 0.641 |
| GADCT [22] | 0.767 | 38.99 | 32.29 | 0.768 | 0.697 |
| CNN [23] | 0.768 | 40.67 | 31.49 | 0.796 | 0.605 |
| VGG-19[24] | 0.821 | 40.89 | 33.31 | 0.780 | 0.792 |
| SNN+VGG-19 (Proposed) | **0.933** | **33.02** | **35.44** | **0.820** | **0.840** |

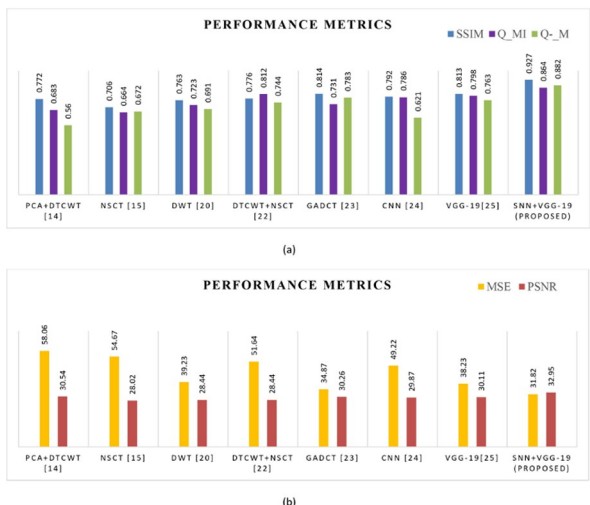

**Fig 13.** (a) & (b) Comparative analysis of objective metrics with other algorithms for the fusion of CT and PET images.

Here the Performance metrics have been compared with popular fusion methods, such as NSCT, DTCWT+NSCT, PCA+DTCWT, and with some pre-trained models like CNN, and VGG-19. The hybrid deep learning technique's performance metrics evaluations yield superior outcomes than those of other methods presently in use. The suggested approach boosts spatial detail information while preserving edge information with the use of objective criteria analysis. Consequently, the multimodal medical image fusion approach that has been proposed has been effective in achieving both the subjective and

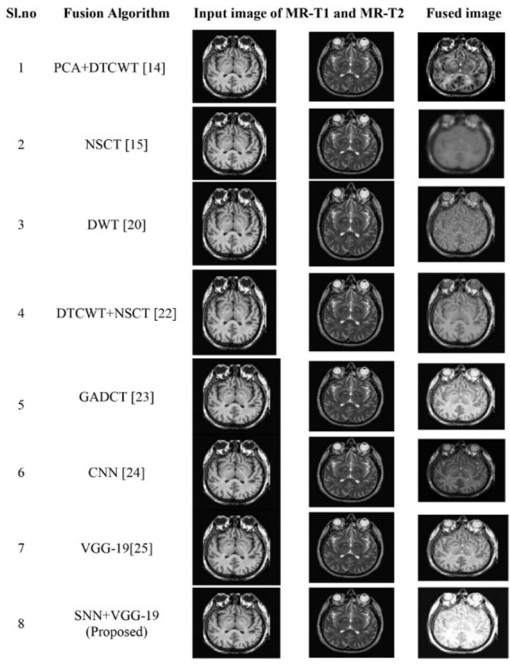

**Fig 14. Comparative analysis of proposed model with various existing fusion models for fusion of MR-T1 and MR-T2 images.**

**Table 5. Resultant objective metrics of fused MR T1-MR T2 of proposed model in comparison with for various existing fusion algorithms.**

| Fusion Algorithms | PERFORMANCE METRICS | | | | |
|---|---|---|---|---|---|
| | SSIM | MSE | PSNR | Q_MI | Q_M |
| PCA+DTCWT [13] | 0.633 | 55.43 | 30.21 | 0.690 | 0.432 |
| NSCT [14] | 0.549 | 59.67 | 29.80 | 0.609 | 0.486 |
| DWT [19] | 0.618 | 32.11 | 30.545 | 0.764 | 0.731 |
| DTCWT+NSCT [21] | 0.618 | 50.13 | 30.54 | 0.779 | 0.467 |
| GADCT [22] | 0.566 | 38.66 | 30.075 | 0.723 | 0.590 |
| CNN [23] | 0.782 | 48.45 | 31.69 | 0.764 | 0.613 |
| VGG-19 [24] | 0.883 | 34.66 | 32.48 | 0.803 | 0.547 |
| SNN+VGG-19 (Proposed) | **0.927** | **29.90** | **34.36** | **0.844** | **0.862** |

objective criteria for assessment. The experimental findings for various fusion methods in comparison to the proposed model are shown in Figs 9, 11, 13 and 15, in tandem with the corresponding performance metrics provided in Tables 3 to 6. Take note that the higher performance value in each of Tables 3 to 6's columns is bolded and that Tables 3 to 6's graphs are presented in Figs 10, 12, 14 and 16 (a & b). For every performance metric, the proposed approach significantly performs better than the existing methods.

## 6. Limitations and future work recommendations

Combining two powerful architectures can be computationally expensive, and the model limits its fusion operation to only two source images. In the future, we will continue to explore the great potential of deep neural networks or ensembles [40] and apply them to other types, such as infrared-visible and multi-focus image fusion. We intended to extend our work for tri-modal image fusion also. Various combinations of deep learning architecture will be carried out to reach more robust fusion operations. The generated fused output can be considered as a source image for segmentation and classification tasks in the effective detection and diagnosis of many tumors.

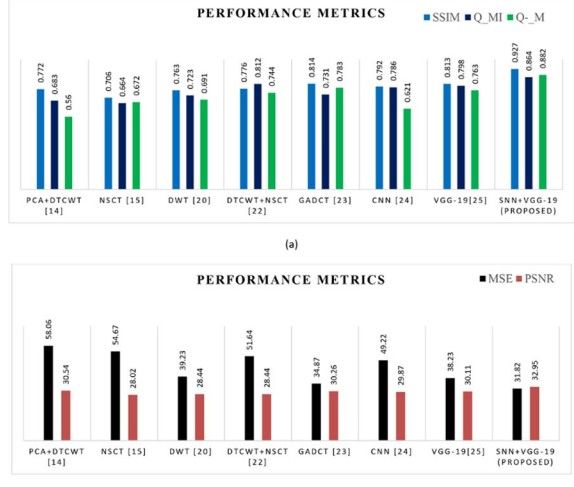

**Fig 15.** (a) & (b) Comparative analysis of objective metrics with other algorithms for the fusion of MR-T1 & MR-T2.

**Table 6. Resultant objective metrics of fused CT-MRI of proposed model in comparison with for various existing fusion algorithms.**

| Fusion Algorithms | PERFORMANCE METRICS | | | | |
|---|---|---|---|---|---|
| | SSIM | MSE | PSNR | Q_MI | Q_M |
| PCA+DTCWT [13] | 0.791 | 58.06 | 30.54 | 0.645 | 0.562 |
| NSCT [14] | 0.763 | 54.67 | 28.02 | 0.673 | 0.434 |
| DWT [19] | 0.704 | 39.23 | 28.44 | 0.771 | 0.534 |
| DTCWT+NSCT [21] | 0.792 | 51.64 | 28.44 | 0.801 | 0.486 |
| GADCT [22] | 0.785 | 34.87 | 30.26 | 0.834 | 0.683 |
| CNN [23] | 0.801 | 49.22 | 29.87 | 0.819 | 0.498 |
| VGG-19 [24] | 0.815 | 38.23 | 30.11 | 0.834 | 0.623 |
| SNN+VGG-19 (Proposed) | **0.947** | **31.82** | **32.95** | **0.852** | **0.790** |

## 7. Conclusions

Considering the growing significance of deep learning in medical image applications, in this paper we proposed a DL model with the use of SNN in combination with VGG-19 to perform image fusion. We utilize the capabilities of trained networks of CNN and VGG-19 to identify and extract deep feature maps that describe the most significant regions in images. Furthermore, we demonstrate that our approach produces a high-quality fused image with improved imaging quality for various combinations of source images. The superiority of model performance is evaluated both on visual and quantitative metrics, subsequently indicating that it is the most effective tool for radiologists to make use of effective and precise medical image analysis.

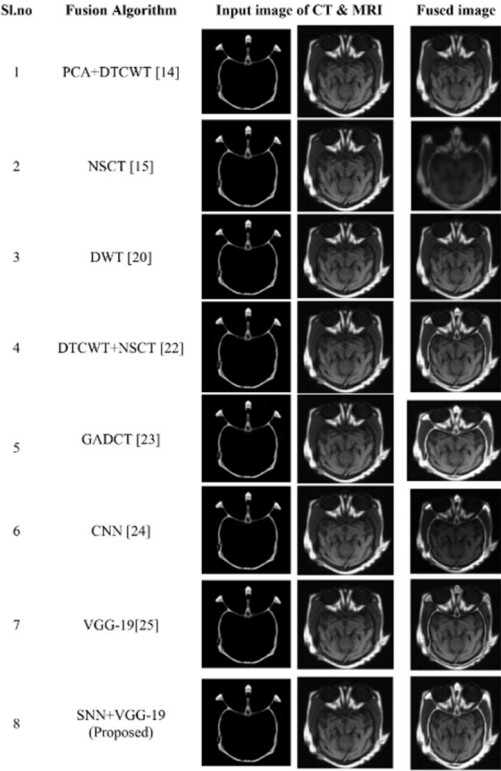

**Fig 16. Comparative analysis of proposed model with various existing fusion models for fusion of CT and MRI images.**

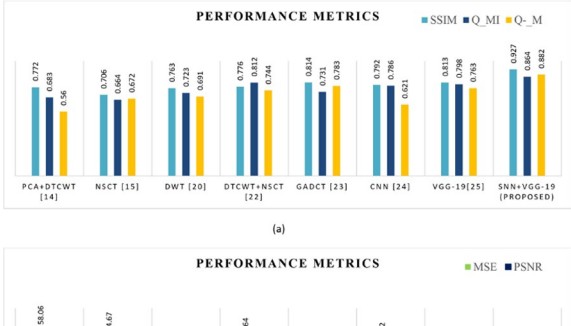

**Fig 17.** (a) & (b) Comparative analysis of objective metrics with other algorithms for the fusion CT & MRI image.

## Author Contributions

**Conceptualization:** Venu Allapakam.

**Data curation:** Venu Allapakam.

**Formal analysis:** Venu Allapakam.

**Methodology:** Venu Allapakam.

**Resources:** Venu Allapakam.

**Supervision:** Yepuganti Karuna.

**Validation:** Venu Allapakam, Yepuganti Karuna.

**Writing – original draft:** Venu Allapakam.

**Writing – review & editing:** Yepuganti Karuna.

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
