## [Decision Letter · Decision Letter 0]

5 Mar 2024

PONE-D-24-04402An Ensemble Learning Model Based on Siamese Neural Networks and VGG-19 for Medical Image FusionPLOS ONE

Dear Dr. KARUNA,

Thank you for submitting your manuscript to PLOS ONE. After careful consideration, we feel that it has merit but does not fully meet PLOS ONE’s publication criteria as it currently stands. Therefore, we invite you to submit a revised version of the manuscript that addresses the points raised during the review process.

We look forward to receiving your revised manuscript.

Kind regards,

Anas Bilal, Ph.D.

Academic Editor

PLOS ONE

Journal Requirements:

Reviewers' comments:

Reviewer's Responses to Questions

**Comments to the Author**

1. Is the manuscript technically sound, and do the data support the conclusions?

Reviewer #1: No

Reviewer #2: Yes

Reviewer #3: Partly

Reviewer #4: Partly

Reviewer #5: Partly

2. Has the statistical analysis been performed appropriately and rigorously? 

Reviewer #1: No

Reviewer #2: Yes

Reviewer #3: No

Reviewer #4: No

Reviewer #5: No

3. Have the authors made all data underlying the findings in their manuscript fully available?

Reviewer #1: Yes

Reviewer #2: Yes

Reviewer #3: Yes

Reviewer #4: Yes

Reviewer #5: No

4. Is the manuscript presented in an intelligible fashion and written in standard English?

Reviewer #1: No

Reviewer #2: Yes

Reviewer #3: No

Reviewer #4: Yes

Reviewer #5: Yes

5. Review Comments to the Author

Reviewer #1: Does the study present an entirely innovative methodology for integrating medical images, or does it merely amalgamate pre-existing techniques? Justify.

Have the outcomes undergone a thorough process of validation using a representative dataset and suitable metrics? Justify.

In what ways does the suggested approach enhance patient outcomes and be applicable to real-world clinical scenarios?

Add more of the latest papers to the literature section:

https://iopscience.iop.org/article/10.1088/1742-6596/1478/1/012024/meta

https://link.springer.com/article/10.1007/s13721-021-00342-2

https://www.taylorfrancis.com/chapters/edit/10.1201/9781003333081-4/medical-image-fusion-rajalingam-santhoshkumar-deepan-santosh-kumar-patra

https://www.sciencedirect.com/science/article/pii/S1746809423009679

https://www.sciencedirect.com/science/article/pii/S1746809421003852

Does the approach acknowledge its limitations and engage in a discussion regarding potential avenues for improvement?

The lack of mention in the title regarding the particular type of ensemble learning implemented (e.g., stacking, boosting) may give rise to inquiries regarding the approach's originality and efficacy. More detail is required.

Reviewer #2: Does the proposed model offer a unique approach to medical image fusion compared to existing methods? Remark it.

Is the methodology clearly described and well-executed? Are the chosen architectures (Siamese networks, VGG-19) appropriate for the task? Justify.

Does the model achieve state-of-the-art results on relevant benchmark datasets? Are the metrics used for evaluation appropriate and informative?

Literature is not upto the mark include recent literature such as Multimodality Medical Image Fusion Using Clustered Dictionary Learning in Non-Subsampled Shearlet Transform; A Non-Conventional Review on Multi-Modality-Based Medical Image Fusion; Directive clustering contrast-based multi-modality medical image fusion for smart healthcare system; Multi-modal medical image fusion in NSST domain for internet of medical things; Multimodal medical image fusion in NSST domain with structural and spectral features enhancement

How well does the model perform on unseen data? Are there limitations to its generalizability?

Is the dataset used for training and evaluation sufficiently large and diverse? Are there potential biases in the data?

Is the model computationally efficient and feasible for real-world applications?

Can the model explain its decisions and highlight the most important features in the fused image?

Does the model provide clinically meaningful insights? How would it be integrated into the clinical workflow?

Reviewer #3: The paper is missing of concise ordering of ideas. It is weak in describing sound research methods, statistical approaches, and interpretation. The study limitations have not been included. I would like to recommend the following points to improve the article:

1. The abstract is adequate in length and structure. The availability of the dataset used in this research effort should be highlighted in the abstract along with the salient results. The comparison with SOTA techniques should be mentioned in the abstract.

2. The main innovation and contribution of this research should be clarified in the introduction. The introduction needs massive improvement to get it in an acceptable form. The motivation of the work is not clear in the introduction.

3. The proposed framework needs improvement. Figures 1 and 2 are too simple to be considered.

4. Figure 3 is weakly representing the proposed method and seems in contradiction to Figures 1 and 2. The dimensional stability in the figures is also not clear due to the absence of details.

5. Thoroughly check your article for typos and grammatical mistakes.

6. The alignment procedure needs more text for explanation. How tie points are aligned in images of both types?

7. Figure 4 is missing of HL and LH components of wavelet transform-based decomposition.

8. All the figures are of poor quality and need improvement to meet the Journal standards.

9. Section 3.1 is missing data availability/address details.

10. In Section 3.2, you have referred to almost all the available methods for preprocessing. Add a flowchart for these methods employed in your work with proper citations.

11. I am not satisfied with Section 3.3. You need to show how you invoked the process.

12. The lack of uniformity of format has been critically observed in the article.

13. Section 3.4 is lacking of technical details.

14. I am not convinced with Figure 6 which is blurred and carelessly drawn. Table 1 depicts the wrong configuration. You need to revise it.

15. I am not satisfied with the results and analysis part of the work. It needs thorough revision.

16. Add a subsection as “Limitations and future work recommendations” before the conclusions. Add the references in the context of future work guidelines:

https://www.nature.com/articles/s41598-023-30309-4

https://doi.org/10.1016/j.inffus.2019.12.012

17. The conclusions need thorough revision.

Reviewer #4: The author proposes an ensemble learning-based image fusion model using a Siamese neural network (SNN) and VGG-19 and achieves specific results in PSNR and SSIM metrics.

(1)The figures presented in the paper are unclear; therefore, it is recommended that vector graphics with a resolution of 600 dpi be used to enhance their clarity and quality.

(2)The paper's contribution is relatively weak, lacking in innovation, and numerous similar methods were developed several years ago.

(3)The methods used for comparison are all traditional multi-scale decomposition methods, lacking comparison with related deep learning-based methods.

(4)The paper only compares based on the objective evaluation metrics of PSNR and SSIM, which cannot demonstrate the effectiveness of the proposed method. Please add corresponding evaluation metrics.

(5)The experimental section of the author's work is deficient in relevant ablation experiments, such as exploring various fusion rules for the proposed method. This would enhance the rigor and comprehensiveness of the study.

Reviewer #5: The article is suitable for publication, after the author addresses and incorporates the following in the manuscript

1. How are the components of the medical images decomposed using the Dual Tree Complex Wavelet Transformation (DTCWT)-based multi-scale image decomposition?

2. How are these fusion rules determined, and what criteria are used to select the most appropriate rule for each component?

3. How does the model perform active level measurement and fusion rules jointly in a single step?

4. How does the model generalize to unseen datasets or different combinations of source images?

5. Can you explain in more detail how the weight map is generated for the base information in the fusion process?

6. What factors or characteristics of the source images are considered in determining the weight values for each pixel?

7. Are there any specific challenges or complexities involved in calculating the weight map, especially across multiple source images with different modalities?

8. Are there any considerations or adjustments made to account for differences in saliency between different types of medical images?

9. Can you explain the significance of the weighted sum pixel calculation equation in the fusion process?

10. How do the weighting factors (α1 and α2) affect the contribution of each source image to the fused base part?

11. Are there any considerations or trade-offs involved in selecting the values of α1 and α2, such as preserving common features and reducing redundant information?

12. Can you provide insights into how the fused base part captures and integrates information from both source images while preserving common features?

13. What techniques or approaches are used to optimize the parameters of the Siamese network and pre-trained CNN model for weight map generation?

14. Required a detailed ablation study

6. PLOS authors have the option to publish the peer review history of their article (what does this mean?). If published, this will include your full peer review and any attached files.

Reviewer #1: No

Reviewer #2: No

Reviewer #3: **Yes: **Prof Dr Shahzad Ahmad Qureshi

Reviewer #4: No

Reviewer #5: **Yes: **Dr. Laavanya Mohan

---

## [Author Response · Author response to Decision Letter 0]

23 Apr 2024

All the comments are addressed in the revised manuscript.

---

## [Decision Letter · Decision Letter 1]

8 May 2024

PONE-D-24-04402R1An Ensemble deep Learning Model for Medical Image Fusion with Siamese Neural Networks and VGG-19PLOS ONE

Dear Dr. KARUNA,

Thank you for submitting your manuscript to PLOS ONE. After careful consideration, we feel that it has merit but does not fully meet PLOS ONE’s publication criteria as it currently stands. Therefore, we invite you to submit a revised version of the manuscript that addresses the points raised during the review process.

We look forward to receiving your revised manuscript.

Kind regards,

Anas Bilal, Ph.D.

Academic Editor

PLOS ONE

Reviewers' comments:

Reviewer's Responses to Questions

**Comments to the Author**

1. If the authors have adequately addressed your comments raised in a previous round of review and you feel that this manuscript is now acceptable for publication, you may indicate that here to bypass the “Comments to the Author” section, enter your conflict of interest statement in the “Confidential to Editor” section, and submit your "Accept" recommendation.

Reviewer #1: All comments have been addressed

Reviewer #3: (No Response)

Reviewer #4: (No Response)

Reviewer #5: All comments have been addressed

2. Is the manuscript technically sound, and do the data support the conclusions?

Reviewer #1: Yes

Reviewer #3: Partly

Reviewer #4: Yes

Reviewer #5: Partly

3. Has the statistical analysis been performed appropriately and rigorously? 

Reviewer #1: Yes

Reviewer #3: No

Reviewer #4: N/A

Reviewer #5: Yes

4. Have the authors made all data underlying the findings in their manuscript fully available?

Reviewer #1: Yes

Reviewer #3: Yes

Reviewer #4: Yes

Reviewer #5: Yes

5. Is the manuscript presented in an intelligible fashion and written in standard English?

Reviewer #1: Yes

Reviewer #3: No

Reviewer #4: Yes

Reviewer #5: Yes

6. Review Comments to the Author

Reviewer #1: The paper can be accepted in the current form, as authors have incorporated all the comments in the paper in proper way.

Reviewer #3: My suggestions have not been addressed.

I ask the authors again to carefully address my points by informing me where the changes could be found in the revision.

Reviewer #4: （1）The author consistently emphasizes that the proposed method is founded upon a deep learning-based image fusion approach. However, the main framework of the proposed method is still based on traditional multi-scale decomposition, with the introduction of VGG-19 and SNN for feature extraction and fusion in the fusion rule setting. I believe that this method still belongs to the category of traditional multi-scale fusion methods (requiring manual parameter adjustment, not end-to-end), and it is not appropriate for the author to claim that the proposed method is based on DL.

（2）The manuscript still lacks corresponding ablation experiments, such as the number of decomposition levels and important parameters of filters in DTCWT, as well as the effectiveness ablation experiments of VGG-19 and SNN modules.

Reviewer #5: The manuscript comprehensively addresses the fusion methodology and contributing valuable insights to the field. All the suggestions are addressed by the authors. I accept the manuscript for publication.

7. PLOS authors have the option to publish the peer review history of their article (what does this mean?). If published, this will include your full peer review and any attached files.

Reviewer #1: No

Reviewer #3: **Yes: **Prof. Dr. Shahzad Ahmad Qureshi

Reviewer #4: No

Reviewer #5: **Yes: **Dr. M. Laavanya

---

## [Author Response · Author response to Decision Letter 1]

24 May 2024

All the reviewer comments are addressed in the revised manuscript.

---

## [Decision Letter · Decision Letter 2]

16 Aug 2024

An Ensemble deep Learning Model for Medical Image Fusion with Siamese Neural Networks and VGG-19

PONE-D-24-04402R2

Dear Dr. KARUNA,

I sincerely apologise for the unusually delayed review timeframe for your revised manuscript. It has now been evaluated by two of the original reviewers, whose comments are appended below. We’re pleased to inform you that your manuscript has been judged scientifically suitable for publication and will be formally accepted for publication once it meets all outstanding technical requirements.

Kind regards,

Emily Chenette, PhD

Editor in Chief

PLOS ONE

Additional Editor Comments (optional):

Reviewers' comments:

Reviewer's Responses to Questions

**Comments to the Author**

1. If the authors have adequately addressed your comments raised in a previous round of review and you feel that this manuscript is now acceptable for publication, you may indicate that here to bypass the “Comments to the Author” section, enter your conflict of interest statement in the “Confidential to Editor” section, and submit your "Accept" recommendation.

Reviewer #3: (No Response)

Reviewer #4: All comments have been addressed

2. Is the manuscript technically sound, and do the data support the conclusions?

Reviewer #3: No

Reviewer #4: (No Response)

3. Has the statistical analysis been performed appropriately and rigorously? 

Reviewer #3: No

Reviewer #4: (No Response)

4. Have the authors made all data underlying the findings in their manuscript fully available?

Reviewer #3: Yes

Reviewer #4: (No Response)

5. Is the manuscript presented in an intelligible fashion and written in standard English?

Reviewer #3: No

Reviewer #4: (No Response)

6. Review Comments to the Author

Reviewer #3: All my points have not been addressed carefully, may be due to the access of reviewers. A better idea is to have a limited number of reviewers, so that the authors can address thoroughly each of them.

Reviewer #4: (No Response)

7. PLOS authors have the option to publish the peer review history of their article (what does this mean?). If published, this will include your full peer review and any attached files.

Reviewer #3: **Yes: **Prof Dr Shahzad Ahmad Qureshi

Reviewer #4: No

---

## [Editor Report · Acceptance letter]

22 Aug 2024

PONE-D-24-04402R2 

PLOS ONE

Dear Dr. KARUNA, 

I'm pleased to inform you that your manuscript has been deemed suitable for publication in PLOS ONE. Congratulations! Your manuscript is now being handed over to our production team.

Kind regards, 

on behalf of

Dr Emily Jane Chenette 

Staff Editor

PLOS ONE